# Clinical Effectiveness and Adverse Events of Bee Venom Therapy: A Systematic Review of Randomized Controlled Trials

**DOI:** 10.3390/toxins12090558

**Published:** 2020-08-29

**Authors:** Soobin Jang, Kyeong Han Kim

**Affiliations:** 1Clinical Medicine Division, Korea Institute of Oriental Medicine, Daejeon 34054 Korea; suebin@nate.com; 2Department of Preventive Medicine, College of Korean Medicine, Woosuk University, Jeonju 54986, Jeollabuk-do, Korea

**Keywords:** bee venom, bee venom therapy, bee venom acupuncture, bee sting, efficacy, safety

## Abstract

Bee venom has been used to treat many diseases because of its anti-inflammatory and analgesic effects. However, the secretions of bee venom can also cause life-threatening adverse reactions. The objective of this paper was to review the clinical effectiveness of bee venom and adverse events induced by bee venom, regardless of the disease. Four electronic databases were searched in April 2020. The reference lists of the retrieved articles and previous review articles were also hand-searched. Randomized controlled trials (RCTs) using any type of bee venom other than live bee stings for the clinical treatment of any disease other than cancer were included. The studies were selected, the data were extracted, and the quality of the studies was assessed by two authors. Risk of bias was assessed using the Cochrane risk of bias standards. Twelve RCTs were included in this review—three on Parkinson’s disease, four on arthralgia, four on musculoskeletal disorders, and one on polycystic ovary syndrome. The types of bee venom used were acupuncture injections, ultrasound gel, and an ointment. Six studies reported adverse events, and skin reactions such as pruritus and swelling were the most common. The large-scale clinical trials of bee venom therapy are needed to verify the statistical difference, and the reporting system for adverse events is also required to increase the safety of bee venom therapy.

## 1. Introduction

Bee venom is secreted from worker bees and is one of the most well-known animal venoms. It consists of mast cell degranulation peptide, melittin, histamine, phospholipase A2 (PLA2), hyaluronidase, acid phosphatase, and nonpeptide components such as glucose and fructose [1]. In general, the pH of bee venom is 5.2 to 5.5, which indicates acidity, and its specific gravity is 1.313 [2]. When bee venom enters the human body, various chemical agents of bee venom cause allergic reactions and, in severe cases, anaphylactic shock as a hypersensitivity [3,4]. PLA2 is the enzyme that is the major allergen in bee venom, and PLA2 secreted from bee venom is associated with inflammation and pain [5]. Bee venom derived PLA2.

Despite its toxicity, bee venom has also been utilized for therapeutic purposes in many clinical cases [6,7]. Bee venom therapy (BVT) is used for various diseases, and in particular, it is known to be effective for musculoskeletal diseases, including arthritis, arthralgia and immune-related diseases, because bee venom has anti-inflammatory and analgesic effects [8,9]. According to animal studies, bee venom has pharmacological effects, including the inhibition of cyclo-oxygenase-2, expression of PLA2 and reduction of tumor necrosis factor α, interleukin (IL)-1, IL-6, nitric oxide, and oxygen-reactive species [10,11].

There are several types of BVT—live bee stings, bee venom acupuncture (BVA) or injections, and externally applied bee venom ointments. Live bee stings, a traditional method of BVT, are applied directly on human skin; therefore, they have a high risk of adverse reactions. BVA is applied by injecting bee venom diluted to a ratio of less than 1:10,000 to minimize the side effects while providing therapeutic effects [2]. A major difference between BVA and bee venom injections is that BVA is administered on the basis of acupoints.

According to a previous review of adverse reactions from bee venom [12], 184 adverse events (AEs) were reported among 397 patients who were exposed to bee venom in 20 randomized controlled trials. Bee venom should be carefully evaluated as a medical treatment because it can cause harm as well as medical benefits. Therefore, this systematic review aimed to summarize the clinical effectiveness of bee venom and AEs induced by bee venom, regardless of the disease.

## 2. Methods

### 2.1. Study Registration

The protocol of this review was registered in PROSPERO (registration number CRD42020165821).

### 2.2. Search Method for Identifying the Studies

This study included the following 4 electronic databases—Medline via Pubmed, Embase via Elsevier, the Cochrane Library, and Web of Science. The reference lists of the retrieved articles were hand-searched, and previous review articles were also examined.

The key words used for the search were ‘bee venom’ and ‘bee venom acupuncture’. Only articles written in English and published since 2010 were included. The search was conducted on 6 April 2020.

### 2.3. Inclusion Criteria for This Review

#### 2.3.1. Types of Studies

This systematic review included only randomized controlled trials (RCTs) that were peer-reviewed. Studies with other designs, such as observational studies, cohort studies, case reports, case series, non-RCTs, animal and experimental studies, and theses were excluded.

#### 2.3.2. Types of Participants

Regarding the participants, only those with cancer were excluded. Since chemotherapy can cause several side effects in cancer patients, they were not included in the review of the adverse events of bee venom.

#### 2.3.3. Types of Interventions

All types of bee venom treatments (i.e., bee venom acupuncture, bee venom injections, bee venom cream) were included in this review. Live bee stings and propolis were excluded. RCTs that combined a bee venom intervention with other interventions were also excluded. RCTs in which other interventions (e.g., drugs, massage, and exercise) were administered to all groups in the exact same manner were included.

There were no restrictions on the comparisons. Placebos, active-control groups, no-treatment groups, and wait-list control groups were allowed as control groups.

#### 2.3.4. Type of Outcome Measures

All outcomes reflecting the clinical effectiveness of bee venom were included. However, RCTs that assessed only the treatment rate without any clinical outcomes were excluded from this review. Articles that did not report the measured values before and after treatment were also excluded.

### 2.4. Data Collection, Extraction and Assessment

#### 2.4.1. Selection of Studies

Two authors (SJ and KHK) independently screened the titles and abstracts of the studies retrieved from 4 databases after excluding the duplicate articles. Then, the full texts of the selected articles were reviewed to ensure that each article met the inclusion criteria for this review. When two authors had a difference of opinion, a third reviewer resolved the disagreement. The entire process is summarized in a flow diagram based on the Preferred Reporting Items for Systematic Reviews and Meta-Analyses (PRISMA) guidelines (Figure 1) [13].

#### 2.4.2. Data Extraction

One author (SJ) extracted the data, and another author (KHK) reviewed the extracted data. Basic information such as the participants, interventions, comparisons, outcomes and adverse events for each included study were recorded.

#### 2.4.3. Assessment of Risk of Bias

Two reviewers (SJ and KHK) assessed the quality of the included studies using the risk of bias (RoB) tool that was developed by Cochrane [14]. RoB was assessed on the basis of the following seven items—(1) random sequence generation; (2) allocation concealment; (3) blinding of participants and personnel; (4) blinding of outcome assessment; (5) completeness of outcome data; (6) completeness of reporting; and (7) other sources of bias. Each item of every RCT was categorized as “high risk (H)”, “unclear (U)”, or “low risk (L)” for all studies. An RoB graph was generated by Review Manager (Cochrane Collaboration Software, RevMan), version 5.3.

## 3. Results

### 3.1. Description of Included Trials

From the four databases that were searched, 370 records were identified, and 322 articles remained after duplicates were removed. After the titles and abstracts were screened, 72 articles remained; after the full-texts were reviewed, 12 RCTs [15,16,17,18,19,20,21,22,23,24,25,26] were included in this review. A flow chart of the study selection process, including the reasons for exclusion, is shown in Figure 1. Among the 12 RCTs, eight were conducted in South Korea [15,17,18,19,21,22,23,25], two in Egypt [20,26], one in France [16], and one in Poland [24]. The number of participants ranged from 20 to 367. Four studies [15,17,18,19] were three-arm clinical trials, and one study [19] included 1-year follow-up results for another study [18]. The treated conditions were Parkinson’s disease, low back pain, temporomandibular disorder (RDC/TMD Ia and RDC/TMD Ib), delayed onset muscle soreness, adhesive capsulitis, pelvic inflammatory disease, knee osteoarthritis, and polycystic ovary syndrome.

The types of bee venom treatments used in the RCTs were acupuncture, ointment, and ultrasound gel. BVA, which is the most typical type of BVT, not only involves stimulation, as does acupuncture, but also involves the injection of bee venom fluids. The bee venom fluid was prepared by dilution in distilled water or NaCl (normal saline). Ointments and ultrasound gel, which are externally applied, were used for the treatment of muscle soreness and temporomandibular disorder.

Placebo, active-control, and no-treatment control groups were included. Normal saline instead of bee venom was injected in the placebo group in six trials [16,17,18,19,22,23], and histamine phosphate was used in one trial [21]. Conventional drugs, acupuncture, Vaseline, and ultrasound without bee venom gel were used in the control groups. The details are described in Table 1.

### 3.2. Risk of Bias of Included Studies

The quality of the included studies was assessed by the RoB tool, which consists of seven areas. Seven RCTs [16,17,18,22,23,24,25] were blinded by using placebo BV, accordingly, the performance bias of those seven studies were assessed as ‘low’. The performance bias of the Yasin et al. study [26] was determined as ‘unclear’ because that study was single-blinded. Since a random sequence with even or odd numbers was generated in the Mohamed et al. study [20], the risk of selection bias of that study was determined to be ‘high’. There were no descriptions about blinding in the follow-up study; therefore, the risk of performance bias and risk of detection bias for Park et al. study [19] were assessed to be ‘unclear’. The risk of attrition bias of the Conrad et al. study [21] was determined to be ‘high’ because of the high drop-out rate. Additionally, a risk of other bias, such as that related to improper funding sources, did not exist. The RoB details are presented in Figure 2.

### 3.3. Effectiveness of the Interventions

#### 3.3.1. Parkinson’s Disease

Among the twelve RCTs included, three RCTs [15,16,17] evaluated the effectiveness of BVA for the treatment of Parkinson’s disease. The BVA group showed a significant improvement in the UPDRS score in two studies, while BVA was not more effective than the placebo in the Hartmanss et al. study [15,16,17]. A qualitative synthesis could not be conducted because different outcome measures were used, and the Cho et al. (2012) [15] study reported median values only and did not report mean values.

#### 3.3.2. Arthritis

Three RCTs on adhesive capsulitis [18], pelvic inflammatory disease [20], and knee osteoarthritis [21] were included, and there was a follow-up study [19] on adhesive capsulitis. The patients with adhesive capsulitis showed improvement in the SPADI score after 12 weeks of treatment and at the 1-year follow-up. The C-reactive protein level and intensity of pain in women with pelvic inflammatory diseases significantly decreased after BV gel was topically applied with an ultrasound device and doxycycline was administered. The WOMAC score in the patients with knee arthritis also improved significantly after bee venom injections.

#### 3.3.3. Musculoskeletal Disorder

There were two trials on low back pain [22,23], one trial [24] on temporomandibular disorder (RDC/TMD Ia and RDC/TMD Ib), and one trial [25] on delayed-onset muscle soreness. BVA decreased the intensity of low back pain but did not significantly increase quality of life in low back pain patients. BV ultrasound gel has been shown to be effective in relieving pain and improving range of motion (ROM) in patients with delayed-onset muscle soreness. Although the Nitecka-Buchta et al. study [24] did not report the *p* value for the comparison between the two groups, BV ointment was more effective than Vaseline in improving muscle tonus and contractions in patients with temporomandibular disorder.

#### 3.3.4. Polycystic Ovary Syndrome

One study [26] evaluated the hormonal changes induced by BV phonophoresis on the low back (BL23) and abdomen (Zigong) combined with a low caloric diet in obese women with polycystic ovary syndrome. The LH/FSH ratio significantly decreased and progesterone level significantly increased after 7 weeks of phonophoresis with topical BV application.

### 3.4. Adverse Events

Adverse events were reported in six RCTs [15,16,18,21,22,23], and there were no safety issues in one study [17]. The remaining five studies did not mention adverse events. The most common symptoms caused by bee venom were skin reactions at the injection sites, including pruritus, rash, and swelling. Systemic symptoms such as headache, nasopharyngitis, and pain in an extremity were also reported. Regarding the meta-analysis results, only itchiness occurred significantly more often in the BV group than in the control group (risk ratio = 6.68, 95% confidence intervals: 2.37, 18.84, *p* < 0.0003, I^2^ = 19%). Detailed information on the adverse events is shown in Table 2 and Figure 3.

## 4. Discussion

BVT is a method of treating diseases that utilizes the pharmacological actions of bee venom, which is used by bees for self-defense, and is widely used in countries worldwide, including China, Korea, Germany, the United Kingdom, Switzerland and France [27]. BVT involving the injection of bee venom at acupoints is called BVA; WHO defined this type of BVT as “a special type of acupuncture performed by bee stings at a certain point or cutaneous region of the meridian/channel for therapeutic purposes, particularly for pain relief” [28]. The venom used for BVA is extracted from the poisons of *Apis Mellifera* using a bee venom collector.

In this systematic review, twelve RCTs were included, and they were divided into three categories on the basis of the diseases treated—Parkinson’s disease, arthritis, and musculoskeletal disorders. Among the three RCTs on Parkinson’s disease, the treatment frequency for the two studies [15,17] showing improvement in the UPDRS score was twice a week. On the other hand, the remaining study [16] showed no effects induced by BVT performed once a month. The frequency of treatment for Parkinson’s disease is recommended to be 2–3 times a week; accordingly, if BVA was administered more frequently in Hartmanss et al.’s study, the results may have been more favorable. The former two studies selected acupuncture points (GB20, LI11, GB34, ST36); however, the latter did not report the injection site. Although there are several preclinical studies [29,30] that have shown the effects of BV on central nervous system diseases such as Parkinson’s disease and multiple sclerosis, the mechanism of BV has not yet been clearly revealed. BVA is assumed to inhibit the development or progression of dysfunction of the central nervous system, such as those observed in patients with PD, multiple sclerosis, and amyotrophic physical sclerosis [30]. However, since the mechanism remains unclear, it is necessary to conduct research on the mechanism as well as on the clinical effects of BVA.

Two studies [18,19] assessed the effects of BVA on adhesive capsulitis; one was a 12-week RCT [18], and the other [19] was a 1-year follow-up RCT. The SPADI scorer significantly improved after 12 weeks of treatment as well as at the one-year follow-up. BVA was also performed for knee osteoarthritis in one RCT, and it was confirmed that various WOMAC scores improved. One RCT reported that BV gel phonophoresis for pelvic inflammation significantly reduced the C-reactive protein level and reduced the pain intensity. BVA, in nonclinical studies, can improve rheumatoid arthritis [31]. Melittin, which is one component of bee venom, has been attracting as an alternative treatment for rheumatoid arthritis due to its anti-inflammatory effects [32]. It also improved degenerative osteoarthritis of the knee joint [33], arthritis of the elbow joint [34], and hip joint disease in case reports and clinical reviews [35].

Generally, arachidonic acid stimulates the production of cyclooxygenase during intracellular metabolic processes and promotes the synthesis of prostaglandins to induce inflammatory reactions [36]. Then, prostaglandin E2 stimulates bradykinin and histamine in afferent nerve terminal receptors, causing sensory hypertrophy with pain and inflammatory reactions in joints, such as edema and vascular dilation [37]. It has been reported that BV inhibits COX-2 and prostaglandin E2 in the body, suggesting that it may be effective in treating inflammation in various parts of a joint [33]. In particular, the treatment effects have been reported in dogs with hip osteoarthritis [38], indicating that this treatment can be applied not only to humans but also to animals. 

The efficacy of BVA in treating low back pain was studied in two RCTs [22,23], both of which showed that BVA significantly reduces pain intensity. There was one RCT [25] that revealed the effect of ultrasound gel with diluted BV on delayed-onset muscle soreness, and the VAS pain score and ROM improved. There was one RCT [24] that showed no significant effects of BV ointment on temporomandibular disorder. BVA involves BV being injected directly into the skin, but BV gel phonophoresis and BV ointment are indirect methods. However, during phonophoresis, transdermal drug delivery is enhanced by ultrasound waves [39]. For ointments, it is difficult for the large-molecular-weight BV particles to penetrate the skin, which may have limited their effects. Therefore, although BV injections may be effective because they are highly absorbent, they cause severe pain [40], and BV phonophoresis [25] is suggested as an alternative. A comparative study of the effects of BV injections and BV phonophoresis should be conducted to determine whether they are safe and effective treatment methods. In particular, the fact that Korean medicine doctors use many pharmacopuncture treatments for musculoskeletal diseases (frequency of 41.7%), neurological diseases (18.0%), and gastrointestinal diseases (9%) indirectly shows that BVA affects musculoskeletal disorders [41].

A total of six RCTs [15,16,18,21,22,23] reported adverse events, which were mostly minor and transient skin reactions, such as pruritus, rash, and swelling. Well-known severe side effects such as anaphylaxis may have been obscured in RCTs with relatively small sample sizes, due to a low incidence of 0.014% [42]. It is also thought that patients with a high risk of hypersensitivity or anaphylaxis were excluded after skin tests and patient histories were performed. In addition, in some studies, granulomas or plaques were observed weeks or months after the BVA treatment [3,43], but in RCTs, the follow-up period is short, so it is less likely to observe these adverse events within the study period.

There are several limitations in this study. First, only those written in English were included in this review. The studies written Chinse or Korean had been omitted. Second, eight of the twelve RCTs included in the study were carried out in Korea, and it is difficult to have sufficient external validity, so this issue should be considered when the results are interpreted or generalized to other populations. Third, it was not discussed that the roles of bee venom compounds in connection to included diseases. This review focused on summarizing the published randomized control trials on the clinical effectiveness of BVT. Fourth, studies targeting cancer patients were excluded although there are references that bee venom has anticancer effects [32]. Since many cancer patients generally get chemotherapy, the authors of this study were concerned that the adverse events of bee venom therapy might be exaggerated. Fourth the number of the included RCTs was not large, and adverse reactions that have a low probability of occurring or are delayed may not have been observed and therefore reported within the short follow-up period.

## 5. Conclusions

This study reviewed the clinical effectiveness of and AEs induced by bee venom treatment. Despite causing mild skin reactions such as pruritus, rash, and swelling, bee venom showed therapeutic effectiveness in treating inflammatory arthritis and musculoskeletal diseases. This study suggests that large-scale clinical trials on bee venom need to be conducted and that a reporting system for AEs needs to be developed to enhance the validity of bee venom treatment.

## Figures and Tables

**Figure 1 toxins-12-00558-f001:**
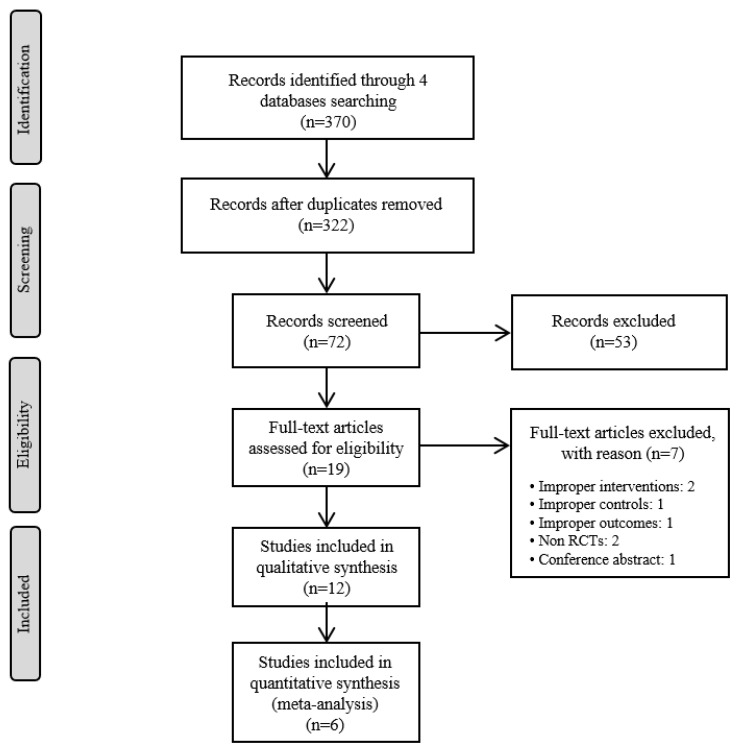
The PRISMA Flow Diagram of Study Selection.

**Figure 2 toxins-12-00558-f002:**
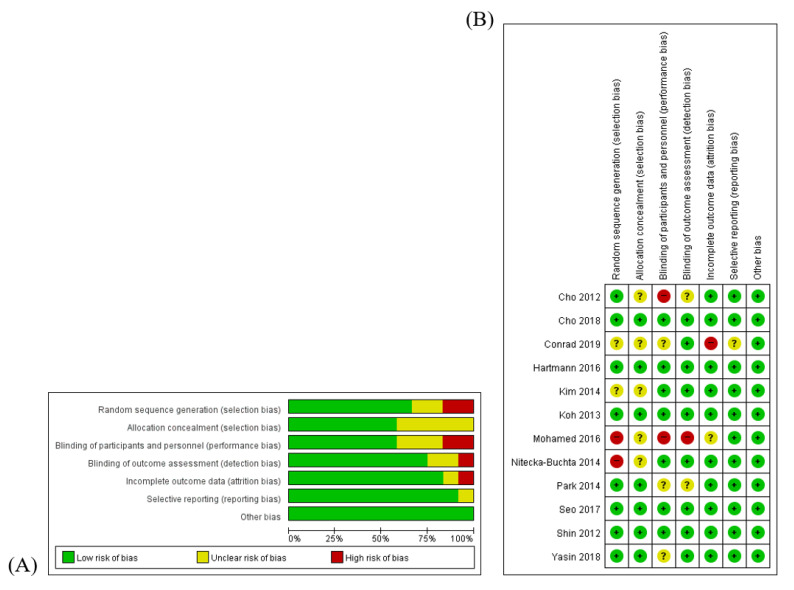
Methodological quality graph. (**A**) Risk of bias graph: authors’ assessments about each risk of bias item presented as percentages of all included studies. (**B**): Risk of bias summary: authors’ assessments about each risk of bias item for each included study. “+”:.

**Figure 3 toxins-12-00558-f003:**
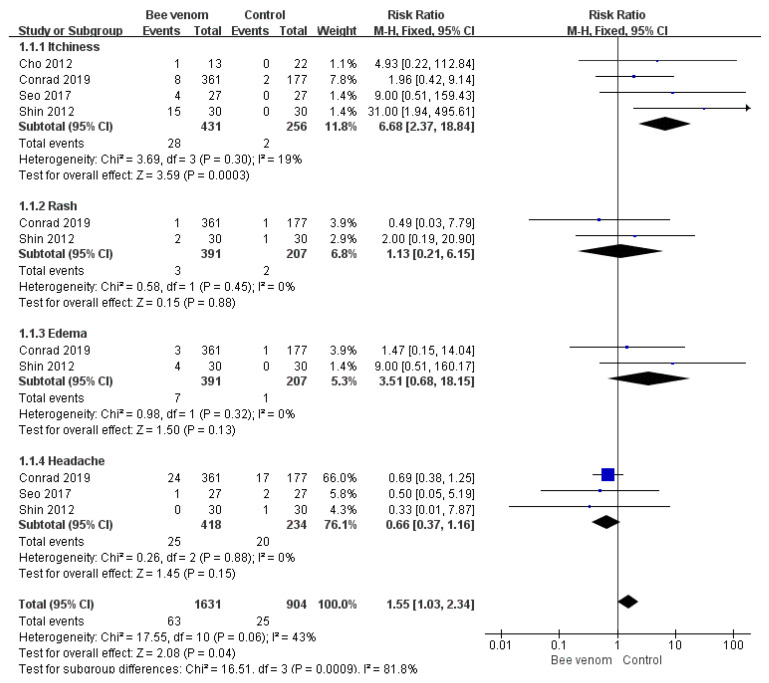
Risk ratio of reported adverse events of bee venom therapy.

**Table 1 toxins-12-00558-t001:** Basic characteristics of included studies.

Condition	Author [Ref] Year	Conducting Country	Age [Mean]	Sex [M/F]	Study Period	Type of Bee Venom	Intervention	Control	Outcome	Results (*p* Value between Two Groups)
Parkinson’s disease	Cho et al. [15] 2012	South Korea	(A) 57.0 (B) 55.0 (C) 57.0	(A) 5/8 (B) 5/8 (C) 3/6	8 weeks	Acupuncture (Injection)	(A) - BVA at bilateral GB20, LI11, GB 34, ST36, and LR3, - BV 0.1 mL diluted on 1:20,000 distilled water - twice a week	(B) - Acupuncture at same points with group A - twice a week (C) No treatment	(1) UPDRS (2) PDQL (3) BDI (4) BBS (5) time of steps required to walk 30 m (6) number of steps required to walk 30 m	[A-B, A-C] (1) NS, *p* < 0.05 (2) NS, NS (3) NS, NS (4) NS, NS (5) NS, NS (6) NS, NS
	Hartmann et al. [16] 2016	France	(A) 60.3 (median) (B) 63.3 (median)	(A) 8/12 (B) 12/8	11 months	Injection	(A) - BV 0.1mg diluted in 1 mL of NaCl 0.9% - once a month	(B) - Placebo (normal saline 1 mL) - once a month	(1) UPDRS (2) Hoenh and Yahr stages (3) ADL (4) BREF (5) MMS (6) PDQ-39 (7) [123I]-FP-CIT binding potential	(1) NS (2) NS (3) NS (4) NS (5) NS (6) NS (7) NS
	Cho et al. [17] 2018	South Korea	(A) 64.4 (B) 61.3 (C) 64.1	(A) 14/10 (B) 8/16 (C) 10/5	12 weeks + 8 weeks (follow-up)	Acupuncture (Injection)	(A) - BVA at bilateral GB20, LI11, GB 34, ST36, and LR3 - dried BV 1 mg diluted in 20 mL of normal saline - twice a week	(B) - Placebo (normal saline injection at same points with group A - twice a week (C) Antiparkinsonian drugs	(1) UPDRS (2) PDQL (3) BDI (4) PIGD (5) Gait speed (6) Gait number (7) MXE (8) DCL	[A-B, A-C] (1) NS, *p* = 0.001 (2) NS, *p* = 0.968 (3) NS, NS (4) NS, *p* = 0.001 (5) NS, NS (6) NS, *p* = 0.115 (7) NS, NS (8) NS, NS
Adhesive capsulitis	Koh et al. [18] 2013	South Korea	(A) 55.0 (B) 56.1 (C) 55.1	(A) 6/16 (B) 8/15 (C) 6/17	12 weeks	Acupuncture (Injection)	(A) - BVA 100 cc (1:10,000 of saline) at LI15, LI16, TE14, GB21, C7, SI11, and additional 5 points around the shoulder - Physiotherapy (B) - BVA 300 cc (1:30,000 of saline) at same points with group A - Physiotherapy	(C) - Placebo (normal saline injection) - Physiotherapy	(1) SPADI (2) VAS—at rest, at night, motion (3) ROM—abduction, forward flexion, extension, external rotation	[Among 3 groups] (1) *p* = 0.017 (2) *p* = 0.110 (at rest), *p* = 0.160 (at night), *p* = 0.029 (motion) (3) NS
	Park et al. [19] 2014, follow-up of [18]		(A) 55.4 (B) 52.8 (C) 56.4	(A) 6/16 (B) 8/15 (C) 6/17	1 year				91) SPADI (2) VAS—at rest, at night, motion (3) Recurrence of symptoms	[Among 3 groups](1) *p* = 0.043(2) NS(3) A-1, B-3, C-4
Pelvic inflammatory disease	Mohamed et al. [20] 2016	Egypt	(A) 32.1 (B) 32.5	(A) 0/15 (B) 0/15	4 weeks	Ultrasound gel	(A) - BV gel topical application with phonophoresis - BV 20 μg/gel 1 g - 20 min/time, 3 times per week - doxycycline 100 mg, orally bid for 7 days	(B) Doxycycline 100 mg, orally bid for 7 days	(1) C-reactive protein (2) Pain intensity	(1) *p* < 0.0001 (2) *p* < 0.0001
Knee osteoarthritis	Conrad et al. [21] 2019	South Korea	(A) 56.9 (B) 55.8	(A) 91/143 (B) 64/69	12 weeks	Injection	(A) - BV injection at 2–15 sites escalating over study period (0.1 mL at each site), 10 points at bilateral knees, BL19, BL21, BL23, BL25, and BL27 - dried HV 1.0 mg/1.0 mL 0.5% preservative-free lidocaine	(B) Histamine phosphate injection at same sites with group A	(1-1) WOMAC pain (1-2) WOMAC physical function (2-1) WOMAC VAS resting (2-2) WOMAC VAS walking (3) PGA	(1-1) *p* = 0.001 (1-2) *p* = 0.001 (2-1) *p* = 0.1051 (2-2) *p* = 0.001 (3) *p* = 0.0001
Low back pain	Shin et al. [22] 2012	South Korea	(A) 42.9 (B) 40.0	(A) 13/17 (B) 14/16	4 weeks	Acupuncture (Injection)	(A) - BVA at bilateral BL23, BL24, and BL25 - BV 0.1 mL diluted on 1:2000 distilled water - twice a week	(B) - Placebo (normal saline injection at same points with group A) - twice a week	(1) Pain intensity (VAS) (2) ODQ (3) SF-36	(1) *p* = 0.012 (2) NS (3) NS
	Seo et al. [23] 2017	South Korea	(A) 49.9 (B) 50.1	(A) 9/18 (B) 4/23	3 weeks	Acupuncture (Injection)	(A) - BVA at bilateral BL23, BL24, BL26, and GB30, GV3, GV4, and GV5 - dried BV diluted on 1:20000 normal saline (0.9% NaCl) - Loxonin 60 mg, orally tid for 3 weeks	(B) - Placebo (normal saline injection at same points with group A) - Loxonin 60 mg, orally tid for 3 weeks	(1) Bothersomeness (VAS) (2) Pain intensity (VAS) (3) ODI (4) BDI (5) EQ-5D	(1) *p* = 0.016 (2) *p* = 0.049 (3) *p* = 0.009 (4) *p* = 0.043 (5) *p* = 0.051
Temporomandibular disorder (RDC/TMD Ia and RDC/TMD Ib)	Nitecka-Buchta et al. [24] 2014	Poland	23 (22-34)	(A) 6/28 (B) 4/30	2 weeks	Ointment	(A) - BV ointment (0.012 mg liquid BV) for topical skin application in region of both masseter muscles - Massage	(B) - Vaseline at same region with group A - Massage	(1) Muscle tonus (2) Muscle contraction (3) VAS	(1) NR (2) NR (3) NR
Delayed onset muscle soreness	Kim et al. [25] 2014	South Korea	(A) 27.4 (B) 28.9	(A) 0/10 (B) 0/10	3 days	Ultrasound gel	(A) - Ultrasound gel and diluted bee venom (0.001%) mixed at a ratio of 9:1 - Ultrasound at the belly of the biceps brachii muscle, - 1 MHz, 1.0 W/cm2, 2.5 cm/s, 10 min	(B) - Ultrasound in the same way as group A, with pure ultrasound gel without BV	(1) VAS (2) CK (3) ROM—flexion, extension	(1) *p* < 0.05 (2) NS (3) *p* < 0.05
Polycystic ovary syndrome	Yasin et al. [26] 2019	Egypt	(A) 26.0 (B) 26.3	(A) 0/23 (B) 0/23	14 weeks	Ultrasound gel	(A) - BV gel topical application with phonophoresis at BL23 and Zigong - 0.6–1.0 mg of BV each session, 1 MHz - twice a week - low calorie diet (1200–1400 kcal/day)	(B) - Ultrasound in the same way as group A, with pure ultrasound gel without BV - low calorie diet (1200–1400 kcal/day)	(1) LH (2) FSH (3) LH/FSH (4) Progesterone	(1) *p* = 0.683 (2) *p* = 0.449 (3) *p* = 0.456 (4) *p* = 0.183

BV: Bee Venom; BVA: Bee Venom Acupuncture; UPDRS: Unified Parkinson’s Disease Rating Scale; PDQL: Parkinson’s Disease Quality of Life Questionnaire; BDI: Beck Depression Inventory; BBS: Berg Balance Scale; ADL: Activities of Daily Living ; BREF: Batterie Rapide D’évaluation Frontale ; MMS: Mini Mental State; PDQ-39: Parkinson’s Disease Questionnaire-39; PIGD: Postural Instability and Gait Disorder; MXE: Maximum excursion ; DCL: Directional control VAS: Visual Analog Scale; ODQ: Oswestry Disability Questionnaire; ODI: Oswestry Disability Index; EQ-5D: EuroQol 5-Dimension ; SPADI: Shoulder Pain and Disability Index; ROM: Range Of Motion; CK: Creatine Kinase; WOMAC: Western Ontario and McMaster Universities Arthritis Index; PGA: Patient Global Assessment; NS: Not Significant; NR: Not Reported.

**Table 2 toxins-12-00558-t002:** Reported adverse events of included studies.

Author [Ref] Year	Condition	Type of Bee Venom	Adverse Events [Number of Patients]	
			Bee Venom Group	Control Group
Cho et al. [15] 2012	Parkinson’s disease	Acupuncture	itchiness 1 (drop-out)	-
Hartmann et al. [16] 2016	Parkinson’s disease	Injection	redness/itchiness 165 cases, insomnia 1 case, nausea 3 cases, fatigue 2 cases, dyskinesia 1 case BV specific IgE 18 patients (90%) BV specific IgG4 12 patients (60%)	redness/itchiness 6 cases, insomnia 1, nausea 9 cases, fatigue 10 cases, dyskinesia 1 case, bradycardia 2 cases
Shin et al. [22] 2012	Low back pain	Acupuncture	itchiness 15, skin flare 5, edema 4, rash 2	rash 1, headache 1, hand-foot tingling 1
Seo et al. [23] 2017	Low back pain	Acupuncture	minimal itching sensation (recovered completely without any treatment) 4, headache 1, generalized myalgia 1	headache 2, dizziness 2
Koh et al. [18] 2013	Adhesive capsulitis	Acupuncture	Muller Grade 0 reactions 30, Muller Grade 1 reactions 1	slight redness and pruritus 3
Conrad et al. [21] 2019	Knee osteoarthritis	Injection	swelling 15, discoloration 10, pruritus 8, erythema 7, urticaria 5, hypersensitivity 3, edema 3, injection site pain 3, vesicles 3, rash 1, hematoma 1, paresthesia 1, headache 24, back pain 13, nasopharyngitis 11, upper respiratory tract infection 11, pain in extremity 10, arthralgia 8, diarrhea 10	itchiness 2, erythema 1, urticaria 1, rash 1, headache 17, back pain 9, nasopharyngitis 7, upper respiratory tract infection 4, pain in extremity 4, arthralgia 4, diarrhea 1

## Data Availability

The data used for this study are available from the corresponding author upon request.

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
