# Peer review of "Clinical Effectiveness and Adverse Events of Bee Venom Therapy: A Systematic Review of Randomized Controlled Trials"

_toxins, 2020, doi:10.3390/toxins12090558_

Round 1

Reviewer 1 Report

Comment to authors:

  1. The authors discussed the uses of bee venom therapy, the title should reflect the actual work.
  2. The authors rely on four databases to cover the current topic. The authors are recommended to screen the literature and other database such as pubmed, science direct, …….etc..
  3. “ The key words used for the search were ‘bee venom’ and ‘bee venom acupuncture”, authors are recommended to include topic like safety, clinical trials …etc.
  4. “Regarding the participants, only those with cancer were excluded” why cancer cases were excluded especially bee venom has anticancer properties.
  5. The current work concered on uses bee venom with Parkinson’s disease, arthralgia, musculoskeletal disorders, andovary syndrome. Bee venom was tested also for hepatitis C virus , and Plaque psoriasis, could you explain why you select specific diseases.
  6. “When bee venom enters the human body, ……………., anaphylactic shock [3, 4]” can you describe the mechanism of action corresponding to these diverse effects?
  7. More compound were identified from bee venom. Did you check their roles in connection to these diseases.
  8. Please add abbreviation list or write the full name before the abbreviation the first time it shows up in the manuscript.
  9. Figure 3: The figure caption did not describe the content.
  10. English editing is recommended and the manuscript contains several typoerrors.
  11. The following reference may be of benefit:

Aufschnaiter, A., Kohler, V., Khalifa, S., El-Wahed, A., Du, M., El-Seedi, H. and Büttner, S., 2020. Apitoxin and its components against cancer, neurodegeneration and rheumatoid arthritis: Limitations and possibilities. Toxins12(2), p.66.

Author Response

Thank you for your effort giving comment.

The manuscript was revised as follow. All changes were lighted in yellow.

  1. Title was changed to "Clinical effectiveness and adverse events of bee venom therapy: a systematic review of randomized controlled trials".
  2. The database MEDLINE was changed to Pubmed. The Science direct means Embase.
  3. To broaden the search scope, we have intentionally not included 'safety' or 'clinical trials' in the keyword. It is written in PROSPERO protocol.
  4. It was added as limitation. Thank you for your comment.
  5. This review is the summary of published RCTs regarding on bee venom therapy. RCTs of hepatitis C or plaque psoriasis were not sesarched.
  6. That sentence intended that bee venom can cause hypersensitivity reaction such as anaphylaxis. We think that explaining the biological mechanism of anaphylactic shock is not appropriate to this paper.
  7. It was added as limitation. Thank you.
  8. The full name of the abbreviation was written when it first showed up.
  9. The caption of Figure 3 was written. Thank you.
  10. The manuscript was proofread by AJE; the professional editing company. And, we also checked the typo error. Thank you.
  11. Thank you for introducing good article. It was added as references;#32 of this paper.

Reviewer 2 Report

Submitted for review article (paper No. 856690) entitled “Clinical effectiveness and adverse events of bee venom: a systematic review of randomized controlled trials” is an review paper which shows the clinical effectiveness of bee venom and adverse events induced by bee venom, regardless of the disease. The authors describe the different use of bee venom in various clinical disease. Generally the topic itself is interesting but presented in this way by the authors is not interesting.  All things related to this topic have already been shown in the article from 2019 ,,Bee Venom: Overview of Main Compounds and Bioactivities for Therapeutic Interests’’. The authors try to forcefully present the effects of bee venom in several studies, without explaining too much, so that the reader cannot fully understand the motive of the work. In addition, the review has a typical layout of experimental work divided into introduction, methodology, results and discussions, which significantly undermines the quality of its message. For this reason, I think the work is very synthetic and does not provide any relevant new information on this subject. Besides, there is a lack of explanation of the natural bee venom regarding mechanism of action.

Author Response

This is a systematic review and the explanation on the biological mechanism of bee venom may not be sufficient.

The article you mentioned "Bee Venom: Overview of Main Compounds and Bioactivities for Therapeutic Interests’’ is different with our paper in terms of study design.

Thank you for your effort giving comment.

Reviewer 3 Report

Overall, this is a well-written research report on a rigorously defined set of papers.  As such, it is a useful addition to the literature.  Unfortunately, the greater question of concern relating to bee venom therapy is still poorly resolved, at best, because these types of treatments are either case reports, anecdotal testimonies, or studies in which no realistic controls are available.  One wishes that we would be able to come to solid conclusions relating to the bee venom apitherapy treatment overall, something that has eluded science to this day.

I have just a few minor points for the text:

page 1, line 24 – I would list phospholipase A2 here as it is the most important enzyme in bee venom.  I realize it is discussed later, but think it is worthwhile to mention it at this point in the beginning.

Page 1 line 41 – I would replace "The" with "A major" in the sentence because there are other differences as well, such as BVA injects considerably smaller quantities of venom than BVT.

Page 12, line 60 – the scientific name for the honeybee needs correcting to Apis mellifera in which the species name is in small letters and two ll's are needed.

Author Response

Thank you for your effort giving comment.

All three errors you had pointed were corrected. All changes were lighted in yellow.

Round 2

Reviewer 1 Report

1- Page 12: line 54" "Apis Mellifera" changed to "Apis mellifera"

2- Table 1 & 2 : "Author [Ref]Year " changed to " Authors (year)".